# Establishment of Novel Neuroendocrine Carcinoma Patient-Derived Xenograft Models for Receptor Peptide-Targeted Therapy

**DOI:** 10.3390/cancers14081910

**Published:** 2022-04-10

**Authors:** Catherine G. Tran, Luis C. Borbon, Jacqueline L. Mudd, Ellen Abusada, Solmaz AghaAmiri, Sukhen C. Ghosh, Servando Hernandez Vargas, Guiying Li, Gabriella V. Beyer, Mary McDonough, Rachel Li, Carlos H.F. Chan, Susan A. Walsh, Thaddeus J. Wadas, Thomas O’Dorisio, M Sue O’Dorisio, Ramaswamy Govindan, Paul F. Cliften, Ali Azhdarinia, Andrew M. Bellizzi, Ryan C. Fields, James R. Howe, Po Hien Ear

**Affiliations:** 1Department of Surgery, University of Iowa Carver College of Medicine, Iowa City, IA 52242, USA; catherine-tran@uiowa.edu (C.G.T.); luis-borbon@uiowa.edu (L.C.B.); guiying-li@uiowa.edu (G.L.); gabriella-holmes@uiowa.edu (G.V.B.); marymcdonough23@gmail.com (M.M.); binrui-li@uiowa.edu (R.L.); carlos-chan@uiowa.edu (C.H.F.C.); 2Department of Surgery, Washington University School of Medicine, St. Louis, MO 63110, USA; jmudd@wustl.edu (J.L.M.); rcfields@wustl.edu (R.C.F.); 3Department of Pathology, University of Iowa Carver College of Medicine, Iowa City, IA 52242, USA; ellen-abusada@uiowa.edu (E.A.); andrew-bellizzi@uiowa.edu (A.M.B.); 4The Brown Foundation Institute of Molecular Medicine, McGovern Medical School, The University of Texas Health Science Center at Houston, Houston, TX 77054, USA; solmaz.aghaamiri@uth.tmc.edu (S.A.); sukhen.ghosh@uth.tmc.edu (S.C.G.); servando.hernandezvargas@uth.tmc.edu (S.H.V.); ali.azhdarinia@uth.tmc.edu (A.A.); 5Department of Radiology, University of Iowa Carver College of Medicine, Iowa City, IA 52242, USA; susan-walsh@uiowa.edu (S.A.W.); thaddeus-wadas@uiowa.edu (T.J.W.); 6Department of Internal Medicine, University of Iowa Carver College of Medicine, Iowa City, IA 52242, USA; thomas-odorisio@uiowa.edu; 7Department of Pediatrics, University of Iowa Carver College of Medicine, Iowa City, IA 52242, USA; sue-odorisio@uiowa.edu; 8Department of Medicine, Washington University School of Medicine, St. Louis, MO 63110, USA; rgovindan@wustl.edu; 9Department of Genetics, Washington University School of Medicine, St. Louis, MO 63110, USA; pcliften@wustl.edu

**Keywords:** gastroenteropancreatic neuroendocrine neoplasms, patient-derived xenograft, tumor spheroids, somatostatin receptor-2, near infrared-labelled octreotide analog

## Abstract

**Simple Summary:**

Gastroenteropancreatic neuroendocrine neoplasms (GEP NENs) are a family of rare cancers with rising incidence in recent years. GEP NEN tumor cells are difficult to propagate, and few cellular and patient-derived xenograft (PDX) models are available for testing new therapies and studying the heterogeneous nature of these cancers. Here, we described the establishment and characterization of two novel NEC cellular and PDX models (NEC913 and NEC1452). NEC913 PDX tumors express somatostatin receptor 2 (SSTR2), whereas NEC1452 PDX tumors are SSTR2 negative. As a proof-of-concept study, we demonstrated how these PDX models can be used for peptide imaging experiments targeting SSTR2 using fluorescently labelled octreotide. The NEC913 and NEC1452 PDX lines represent valuable new tools for accelerating the process of drug discovery for GEP NENs.

**Abstract:**

Gastroenteropancreatic neuroendocrine neoplasms (GEP NENs) are rare cancers consisting of neuroendocrine carcinomas (NECs) and neuroendocrine tumors (NETs), which have been increasing in incidence in recent years. Few cell lines and pre-clinical models exist for studying GEP NECs and NETs, limiting the ability to discover novel imaging and treatment modalities. To address this gap, we isolated tumor cells from cryopreserved patient GEP NECs and NETs and injected them into the flanks of immunocompromised mice to establish patient-derived xenograft (PDX) models. Two of six mice developed tumors (NEC913 and NEC1452). Over 80% of NEC913 and NEC1452 tumor cells stained positive for Ki67. NEC913 PDX tumors expressed neuroendocrine markers such as chromogranin A (CgA), synaptophysin (SYP), and somatostatin receptor-2 (SSTR2), whereas NEC1452 PDX tumors did not express SSTR2. Exome sequencing revealed loss of *TP53* and *RB1* in both NEC tumors. To demonstrate an application of these novel NEC PDX models for SSTR2-targeted peptide imaging, the NEC913 and NEC1452 cells were bilaterally injected into mice. Near infrared-labelled octreotide was administered and the fluorescent signal was specifically observed for the NEC913 SSTR2 positive tumors. These 2 GEP NEC PDX models serve as a valuable resource for GEP NEN therapy testing.

## 1. Introduction

Tumors can arise within neuroendocrine cells throughout the body, and some of the most common sites that lead to human morbidity and mortality originate within the GI system, most commonly within the small bowel and pancreas, collectively known as gastroenteropancreatic neuroendocrine neoplasms (GEP NENs). The age-adjusted incidence of GEP NENs increased over sixfold in the United States between 1973 and 2012, with an annual incidence of 3.56 per 100,000 persons [1]. These tumors are typically slow growing, but over 60% of patients present with metastatic disease and survival is greatly diminished [2,3,4]. In addition to the stage of the disease, both tumor grade and cell morphology/degree of differentiation are also powerful predictors of survival. Based on the 2019 World Health Organization classification of neuroendocrine neoplasms, GEP NENs are comprised of well-differentiated neuroendocrine tumors (NETs) and poorly-differentiated neuroendocrine carcinomas (NECs) [5]. Grade 1, 2, and 3 NETs are all morphologically well-differentiated and have Ki-67 values of <3%, 3–20%, and >20%, respectively. NECs are poorly differentiated tumors with a Ki-67 proliferation index >20% and/or a mitotic rate of over 20 mitoses per 2 mm^2^, and many NECs have Ki-67 indices >50% [5]. In addition to these distinguishing features of poor differentiation and high Ki-67, GEP NECs have frequent mutations in *TP53* and *RB1*, while mutations in NET tumors are less common, and include *MEN1*, *DAXX*, and *ATRX* for pancreatic NETs (PNETs) and *CDKN1B* and 18q loss in small bowel NETs (SBNETs) [5,6,7].

Clinically, NECs are very aggressive, rapidly dividing tumors. They are associated with high rates of metastatic disease at presentation and poor prognosis. The incidence of NECs is not as well-defined due to changes in the WHO classification over the past decade, but epidemiological studies estimate this to be about 0.4/100,000 person-years [1,8]. The optimal treatment for GEP NECs has not been established due in part to the rarity of these tumors and difficulty with performing randomized trials. As a result, current guidelines for treatment of GEP NEC are based on lower-level evidence (NCCN Guidelines 2021) [9]. Therapeutic strategies are largely derived from experience in management of small cell lung cancer due to the pathologic and immunohistochemical similarities between small cell lung cancer and GEP NECs [10,11,12]. The appropriateness of this has been called into question as small cell lung cancer differs from NECs in several ways including higher association with smoking, higher rate of brain metastases, and higher sensitivity to platinum-based chemotherapy [11,12,13,14]. Patients are generally treated with chemotherapy regimens including platinum-based alkylating agents (carboplatin) and topoisomerase inhibitors (etoposide). Despite treatment, response rates are only 30–50% and median overall survival is 9 to 20 months [13,14,15].

Current therapies for both NETs and NECs are limited to somatostatin analogues (SSAs), mTOR inhibitors (everolimus), tyrosine kinase inhibitors (sunitinib), limited chemotherapy regimens, and peptide receptor radionuclide therapy (PRRT) [16,17,18]. One of the biggest barriers to identifying additional active therapeutics has been the inability to establish GEP NEN cell lines and mouse models that can be grown robustly. Some of these tumor cells can be grown in culture as spheroids, but they grow very slowly and are difficult to propagate as xenografts [19,20,21]. The two widely used cell lines, BON [22] and QGP1 [23] resemble poorly differentiated NECs [24,25,26], and unfortunately express low levels of NEN markers such as the somatostatin receptor 2 (SSTR2) [24]. Although well-differentiated GEP NET cell lines, such as the P-STS, GOT1, and NT-3 cells, have been described [24,25,27], difficulties growing these cells in abundance have limited their distribution to other researchers. The paucity of available cell lines has been a significant hurdle towards better understanding NEN biology and to provide theranostic models, and therefore we set out to establish new models to expand these options for NEN research.

## 2. Methods

### 2.1. Patient-Derived Xenograft Models and Cell Lines

The inventory of the Washington University PDX center was searched for neuroendocrine tumors and carcinomas, which were collected under an Institutional Review Board-approved protocol (#201708051) of Washington University and cryo-preserved. All peripheral blood and patient tumor tissue were procured on the day of surgery. Peripheral blood was layered onto a Ficoll gradient, peripheral blood mononuclear cells (PBMCs) isolated, rinsed, 5 × 10^6^/mL transferred to cryovials, and cryopreserved in FBS + 10% DMSO. Tumor tissues were kept cold, cut into multiple small pieces, 5 pieces transferred into each cryovial, and cryopreserved in FBS + 10% DMSO. All cryopreserved samples were progressively cooled in a freezing container at a controlled rate of −1 °C/min at −80 °C. Vials were subsequently transferred to liquid nitrogen for long term storage.

Six patient samples were selected for this study. Tumors were thawed, minced, digested with collagenase and DNase I, and strained to obtain a single-cell suspension [19]. One to ten thousand cells were injected subcutaneously into NOD scid gamma (NSG) mice under an Institutional Animal Care and Use Committee-approved protocol (#9051771). Once visible subcutaneous tumors developed, tumor volume was calculated by multiplying tumor length, width, and depth and expressed as mm^3^. Tumors larger than 1000 mm^3^ were harvested, processed, and 1 million tumor cells were injected into another generation of NSG mice, and remaining cells were placed in suspension culture. After 2 days of incubation, mouse fibroblasts from the PDX tumors were easily removed since they adhered to the plastic culture dish while the NEC cells grew in suspension. The floating NEC cells were harvested and transferred into new culture dishes. After 3 passages, the cultures consisted of only NEC cells with no fibroblast contamination was observed. NEC tumor cells can also be seeded in extracellular matrix as 3-dimensional cultures. For optimal NET marker expression, use NEC cells recently isolated from PDX tumors and avoid using cells in culture over 6 months. NEC tumor cells were grown in Dulbecco’s modified Eagle’s medium (DMEM)/F12 with 10% fetal bovine serum (FBS), 1% penicillin/streptomycin, 1% L-glutamine, insulin, and nicotinamide [19]. BON cells were cultured in Dulbecco’s modified Eagle’s medium (DMEM)/F12 with 10% fetal bovine serum (FBS), 1% penicillin/streptomycin, and 1% L-glutamine [22].

### 2.2. Histology and Immunohistochemistry

Patient samples from surgery were fixed in formalin, embedded in paraffin, and sectioned. Slides were deparaffinized, rehydrated, and stained with hematoxylin and eosin (H&E). Slides were immunostained using specific antibodies against chromogranin A (CgA) (Thermo Scientific, Waltham, MA, USA, #MA5-13096), synaptophysin (SYP) (Agilent Dako, Santa Clara, CA, USA, #M7315), achaete-scute family bHLH transcription factor 1 (ASCL1) (BD Pharmingen, San Diego, CA, USA, #556604), p53 (Agilent Dako, #M700101-2), retinoblastoma protein (Rb) (BD Pharmingen, #554136), somatostatin receptor 2 (SSTR2) (Abcam, Waltham, MA, USA, #ab134152), and C-X-C motif chemokine receptor 4 (CXCR4) (Abcam, #ab124824), and Ki-67 (Agilent Dako, #M724001-2). Ki-67 proliferation index was quantified by percentage of positively staining cells in ~500 tumor cells per tumor sample.

### 2.3. Quantitative PCR

RNA was extracted from tumors grown in mice using the RNeasy Plus Universal Kit (Qiagen, Beverly, MA, USA) and reverse transcribed to cDNA using the qScript cDNA Supermix (QuantaBio, Beverly, MA, USA). Quantitative PCR was performed with gene-specific primers and PerfeCTa SYBR Green Supermix dye (Quantabio) using the 7900HT Fast Real-Time PCR System (Applied Biosystems, Waltham, MA, USA). Primer sequences were obtained from PrimerBank (https://pga.mgh.harvard.edu/primerbank, accessed on the 8 July 2020) and were purchased from Integrated DNA Technologies (IDT). Primer sequences used for qPCR analysis are shown in Table 1.

### 2.4. Immunofluorescence

Cells derived from mouse tumors were fixed with 4% paraformaldehyde for 10 min, then stained with primary antibodies against SYP (Abcam, #32127) at 1:600 dilution, CgA (Invitrogen, Waltham, MA, USA, #MA5-13096) at 1:400 dilution, and SSTR2 (Sigma, St. Louis, MO, USA, #HPA007264) at 1:400 dilution overnight. Cells were washed and incubated with an FITC-conjugated secondary antibody (Jackson ImmunoResearch, West Grove, PA, USA, #115-095-062 and #711-095-152) at 1:500 dilution for 1 h. Immunofluorescent images were taken using a fluorescent microscope at 200 ms exposure time.

### 2.5. Genomic DNA Analyses

Genomic DNA from PDX tumors, and peripheral blood mononuclear cells (PBMCs) were extracted using a DNA extraction kit (Qiagen). Short tandem repeats (STR) analyses were performed on the DNA samples using the Cell Check9 panel of 9 human STR polymorphisms (IDEXX, Westbrook, ME, USA). Exome sequencings of PDX tumors were performed by submitting genomic DNA from NEC913 and NEC1452 PDX tumors to the Washington University Genome Technology Access Center for analyses with the IDT Exome 150X coverage. Exome sequencing data were analyzed using a DRAGEN processor and compared to the GRCh38 reference genome.

### 2.6. Imaging of Patient-Derived Xenograft Mouse Model with Bilateral Tumors

Five female NSG mice (Jackson Laboratory, Bar Harbor, ME, USA, Stock no: 005557) were anesthetized with 1% to 2% isoflurane at 10 weeks of age, and were subcutaneously injected with 1 × 10^6^ SSTR2(+) cells in extracellular matrix Matrigel (Corning, Corning, NY, USA, #356235) in the left shoulder, and 1 × 10^6^ SSTR2(−) cells in the right shoulder. When bilateral tumor size reached between 10 to 20 mm in diameter at 5 weeks post-implantation, in vivo and ex vivo near infrared (NIR) fluorescence imaging were conducted with NIR octreotide analog (NIR-TOC) as described by Hernandez-Vargas et al. [28,29] In brief, 6 nmol of NIR-TOC diluted in 100 μL PBS was administered via mouse tail-vein injection 24 h prior to imaging studies. NIR fluorescence imaging was acquired using the IVIS Lumina S5 small animal imaging station and Living Image^®^ software (PerkinElmer, Waltham, MA, USA) with excitation and emission set to 740 and 790 nm, respectively. Images with favorable contrast-to-noise ratio were obtained using exposure time of 2 s for in vivo and 0.1 s for ex vivo imaging, with subject height of 1.50 cm, small binning and F/Stop setting of 2, and field of view setting C. After completing in vivo imaging, mice were euthanized and dissection was immediately performed for ex vivo isolation, and imaging of subcutaneous tumors as well as major intraabdominal and intrathoracic organs was performed. Quantification of NIR fluorescent signal was performed using ImageJ version 1.53 a (NIH, Bethesda, MD, USA). Statistical analyses for NIR fluorescent signal were performed using *t*-tests in Prism GraphPad. *p* < 0.05 was depicted with *.

## 3. Results

Tumor cells from six cryopreserved patient tumors (Table 2) were injected into the flank of NSG mice to generate PDX models (Figure 1A). At 3 months post-tumor cell injection, two mice harboring GEP NEC cells had developed subcutaneous tumors of approximately 1 cm in diameter (NEC913 and NEC1452; Figure 1A) while four GEP NET patient tumor cell samples injected into mice did not form tumors (Table 2). Subcutaneous injection of 1 × 10^6^ NEC913 and NEC1452 cells grew into tumors about 1000 mm^3^ and 1500 mm^3^ in size, respectively, after 5 weeks in subsequent passages. The tumor formation rate was 100%. The NEC913 and NEC1452 xenograft tumors were harvested and collected for histological and biochemical analyses. A separate portion of the NEC913 and NEC1452 tumors was collected for tumor cell isolation and injection into another generation of mice for propagation of the PDX models and for establishment of cell lines. Both NEC913 and NEC1452 cells were successfully maintained in culture for months in enriched DMEM/F12 medium. Both PDX tumors stained positive for the neuroendocrine tumor markers chromogranin A (CgA) and synaptophysin (SYP), but only the NEC913 PDX tumor stained positive for somatostatin receptor 2 (SSTR2; Figure 1B). Exome sequencing of the NEC913 and NEC1452 PDX tumors were performed and mutations in *TP53* and *RB1* were confirmed (Figure 1C; Appendix A).

We were able to retrieve the original patient tumor for the NEC913 sample for comparison with the PDX model. The NEC913 tumor came from a patient presenting with jaundice and upper GI bleeding. A biopsy revealed a Grade 3 NEC of the ampulla of Vater, and the patient was treated with carboplatin/etoposide chemotherapy for 4 months, then had a Whipple procedure, where a 0.3 cm primary tumor with multiple involved nodes were also removed (Figure 2A). Hematoxylin and eosin (H&E) staining showed the presence of both small and large NEC cells (Figure 2B). The NEC913 primary tumor stained positive for Ki-67 in over 80% of cells (Figure 2C). An outside pathology report indicated that the specimen was TTF-1 positive and CDX2 negative. We detected positive staining for CgA, SYP, SSTR2, and ASCL1 (Figure 2D–G). A low level of p53 was detected (Figure 2H); however, exome sequencing data identified several stopgain mutations where the first stopgain mutation is located in codon 147 of *TP53* (Figure 1C, Appendix A) suggesting that the IHC staining detect only the first 146 amino acid fragment of p53. Expression of Rb was lost (Figure 2I). Exome sequencing of the NEC913 PDX tumor revealed an *RB1* frameshift insertion (1091_1092insCG) leading to a premature stop codon (Figure 1C, Appendix A). STR analyses confirmed that the NEC913 patient blood sample shared the same alleles with the NEC913 PDX tumor (Appendix A) and that these samples did not match any of the existing research samples in the IDEXX DSMZ STR database, meaning that they are being reported for the first time.

The patient giving rise to tumor NEC1452 presented with a mediastinal and supraclavicular masses, as well as liver, pancreatic, retroperitoneal, and rectal lesions that were Fluorodeoxyglucose–Positron Emission Tomography (FDG-PET) positive and only mildly DOTA-octreotate (DOTA-TATE) PET avid. A supraclavicular node biopsy showed a small cell NEC and carboplatin/etoposide were started, followed by FOLFIRI after progression, then immunotherapy. A retroperitoneal node was biopsied due to poor response, which showed large cell NEC with a Ki-67 of 80–90%, which the source of this PDX. The treating medical oncologist considered the rectum to be the primary site, because this had the highest FDG-PET avidity, with uptake in perirectal nodes, a nearly obstructing mass seen on sigmoidoscopy, and the presence of *APC* mutations in the tumor (Figure 1C, Appendix A). The NEC1452 PDX tumor sample did not match any pre-existing samples in the IDEXX DSMZ STR database (Appendix A), and exome sequencing confirmed *TP53* stopgain mutations and *RB1* frameshift mutation (Figure 1C, Appendix A).

Tumor cells isolated from the cryopreserved NEC913 and NEC1452 tumors yielded viable cells despite the fact that both patients had been treated with carboplatin/etoposide chemotherapy, suggesting that these NEC cells are resistant to the treatment. Both samples can be robustly passaged as PDXs and tumor cells from the xenografts grow as suspension cultures or as spheroids embedded in extracellular matrix. By immunofluorescent staining, we confirmed the expression of CgA and SYP in both cell lines and SSTR2 in only the NEC913 line (Figure 3A). To further characterize these novel cell lines for additional neuroendocrine cancer markers, gene expression analyses using quantitative PCR was performed (Figure 3B–H). In comparison to the established BON cells, NEC913 was found to have significantly increased *SYP* and *SSTR2* expression (Figure 3B,E). NEC1452 cells were determined to have increased *SSTR1* expression relative to BON cells (Figure 3D).

To demonstrate the utility of NEC PDX models as a potential tool for testing receptor-targeted theranostics, we conducted a proof-of-concept study whereby we established a mouse model with tumor implantations using the NEC913 (SSTR2+) and NEC1452 (SSTR2−) cells in opposite shoulders for SSTR2-targeted imaging (Figure 4A). We then injected these mice with a NIR-TOC, which previously was demonstrated to specifically detects SSTR2 on NEN cells, and imaged them using NIR fluorescence imaging [28,30]. Image analysis revealed that the NIR-fluorescence signal was localized only in the NEC913(SSTR2+) tumor (Figure 4B). To confirm the localization of the NIR fluorescence signal on the NEC tumors, ex vivo NIR fluorescence imaging was performed on both tumors after they were removed from the animals. The NIR fluorescence signal was detected in the NEC913 tumor but not NEC1452 (Figure 4C), corroborating the in vivo imaging results (Figure 4B). Qualitative assessment of SSTR2-mediated uptake was supported by semi-quantitative image analyses, which revealed an approximately twofold increase and a threefold increase in fluorescent signal intensity of NEC913 compared to NEC1452 tumors in the in vivo and ex vivo experiments, respectively (Figure 4D,E).

Further characterization of the NEC913 PDX tumor by IHC showed low expression of p53, which could be due to the specificity of the antibody for the truncated form of p53, and no expression of Rb (Figure 5A), which is similar to the expression pattern observed in original patient tumor IHC analyses (Figure 2H,I). The expression of ASCL1 was lower in the NEC913 PDX tumor (Figure 5A) when compared to the original tumor (Figure 2G). In addition, the NEC913 PDX tumor expressed CXCR4 (Figure 5A). The *CXCR4* expression was also detected in NEC913 spheroids by quantitative PCR and IHC using a specific antibody against CXCR4 (Figure 5B,C).

## 4. Discussion

GEP NETs and NECs are rare cancers with few in vitro and in vivo models available for therapeutic testing [31]. GEP NET cell lines and spheroids are difficult to propagate as they take approximately 2 to 3 weeks to divide [19,22,24,32]. A limited number of GEP NEC cell and organoid lines have been recently published, but distribution remains limited. The two currently available human GEP-NEN-derived cell lines, BON and QGP-1, divide approximately every 3 days and carry *TP53* and *RB1* mutations [25]. They are morphologically poorly differentiated, and have Ki-67 rates that exceed 90%, which define them as NEC cell lines.

Because well-differentiated NETs grow slowly, attempts to propagate them long term have not generally been successful. We have shown that these can be grown in culture for up to 9 months, remained well-differentiated, and expressed NET markers such as synaptophysin, chromogranin, and SSTR2 [19]. However, after about 2 weeks, growth remains fairly constant at a low level. In this study, we were unsuccessful at establishing all four frozen NET samples in immunocompromised mice at 3 months post injection (Table 2). Considering that a majority of GEP NETs are generally slow-growing Grade 2 tumors with a Ki-67 index less than 20%, it is possible that a period longer than 3 months is required for tumor formation. Interestingly, even the SBNET sample SBNET1063 (Table 2), which was a WHO Grade 3 tumor, did not generate a subcutaneous tumor, suggesting that a xenograft model may not be ideal for NET PDX development. There have been a few NET cell lines described such as P-STS, GOT1, NT3 [24,25,27], and a well-differentiated PNET PDX model reported by Chamberlain et al. [33]. Although these appear to be promising GEP NET models, they have not been widely distributed to many investigators. The most likely explanation for this is that they cannot be grown in a large enough quantity to share, or that over time, the cells that do survive could potentially dedifferentiate into NECs.

We established two new NEC cell lines that grow well in culture and can be passaged through several generations of immunocompromised mice. These lines significantly expand the options for study of NECs and were derived from different sites. The NEC913 was derived from an ampullary NEC and NEC1452 from a retroperitoneal node from a patient suspected to have a rectal NEC. Both NEC913 and NEC1452 PDX tumors contain *TP53* and *RB1* mutations that are commonly reported in GEP NECs [6,34,35]. Both lines expressed synaptophysin, chromogranin A, and had Ki-67 of >80%. The NEC913 line expressed SSTR2 and CXCR4 while NEC1452 did not. Tumors from these PDX models can robustly be passaged in more than six generations of mice with a 100% rate of tumor formation. NEC cells from both PDX models can be maintained in culture in media supplemented with insulin and nicotinamide as suspension cultures or spheroids embedded in extracellular matrix.

Several lung NEC lines [36,37] and colon NEC lines such as the HROC47, SS-2, and NEC-DUE1 and 2 [36,38,39,40] have been established, but few pancreas and rectal NEC lines have been reported [39]. Considering the rarity of GEP NEC PDX and cell models, the NEC913 and NEC1452 PDX models developed in this study could be tremendously valuable for a variety of pre-clinical experiments. Here, we showed that the NEC913 line can be useful in confirming the target specificity of NIR-TOC with fluorescence imaging in a clinically relevant model, suggesting high translational potential (Figure 4). NEC913 PDX model maintains high SSTR2 (Figure 1B) and CXCR4 (Figure 5A,B) expression after six generations of passages. The NEC913 cells grew as spheroids in suspension culture or embedded in extracellular matrix, recapitulated characteristics of the PDX tumor, and stained positive for CXCR4 (Figure 5). This cell line also shows great promise as a potential tool for future investigations involving PRRT including testing of combination therapies or highly potent alpha-emitter PRRT [41]. CXCR4 is also emerging as a valuable target in atypical lung carcinoid and small cell lung cancer, and can be targeted with the radiolabeled ligand Pentixafor and Pentixather [42,43]. Thus, NEC913 could also serve as an effective pre-clinical model for PRRT directed at CXCR4. The development and characterization of these NEC913 and NEC452 PDX lines represent valuable new tools that could overcome the significant limitations of existing preclinical models in NEN research and accelerate the process of drug discovery.

## 5. Conclusions

Two novel NEC PDX models (NEC913 and NEC1452) were established from cryopreserved patient tumors. Tumors from these PDX models can robustly be passaged in immunocompromised mice. The NEC913 PDX model maintained SSTR2 expression after six generations of passages and can be visualized using NIR-TOC peptide. In addition, the NEC913 PDX model expressed high level of CXCR4, which makes it potentially useful for CXCR4-targeted theranostics. NEC cells from both PDX models can be maintained in culture in media supplemented with insulin and nicotinamide. Considering the rarity of GEP NEC PDX and cell lines, the NEC913 and NEC1452 PDX models are valuable pre-clinical models for peptide imaging, drug testing experiments, and studying GEP NEC tumor biology.

## Figures and Tables

**Figure 1 cancers-14-01910-f001:**
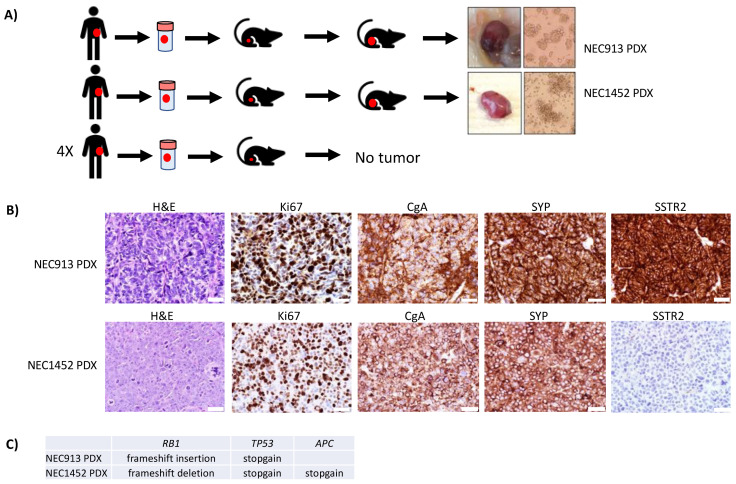
Establishment of neuroendocrine neoplasm (NEN) patient-derived xenograft (PDX) models: (**A**) Tumor samples from NEN patients were cryopreserved, thawed, and injected into the flank of immunocompromised NOD Scid Gamma (NSG) mice. Two mice developed subcutaneous tumors at three months post injection (NEC913 and NEC1452 PDX models). Both PDX models have been passaged in 6 generations of mice. (**B**) Formalin-fixed and paraffin-embedded tumor sections are stained with H&E and stained for Ki67 and neuroendocrine tumor markers such chromogranin A (CgA), synaptophysin (SYP), and somatostatin receptor 2 (SSTR2) by IHC. Scale bar represents 40 μm. (**C**) Exome sequencing of NEC913 and NEC1452 PDX tumors demonstrated mutations in *TP53* and *RB1*.

**Figure 2 cancers-14-01910-f002:**
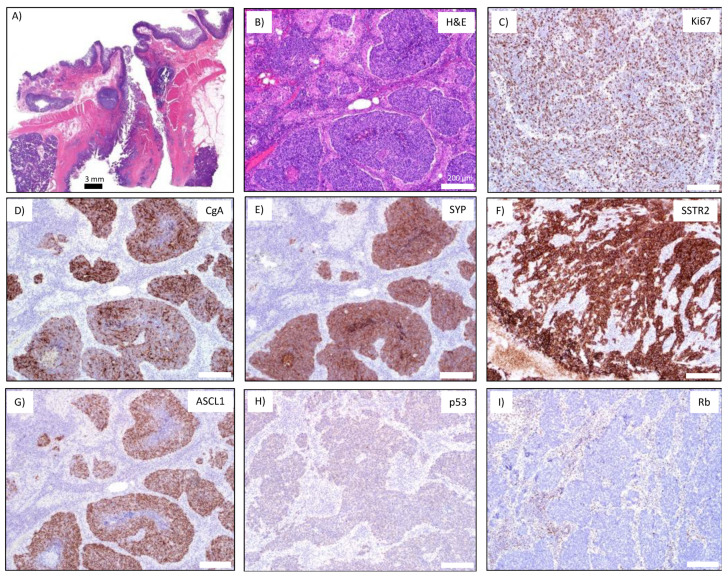
IHC analyses of NEC913 patient sample: (**A**) Primary NEC tumor at the ampulla of Vater. Scale bar represents 3 mm. (**B**) H&E staining of primary NEC tumor. (**C**–**I**) Staining for Ki67, CgA, SYP, SSTR2, ASCL1, p53, and Rb. Scale bar represents 200 μm.

**Figure 3 cancers-14-01910-f003:**
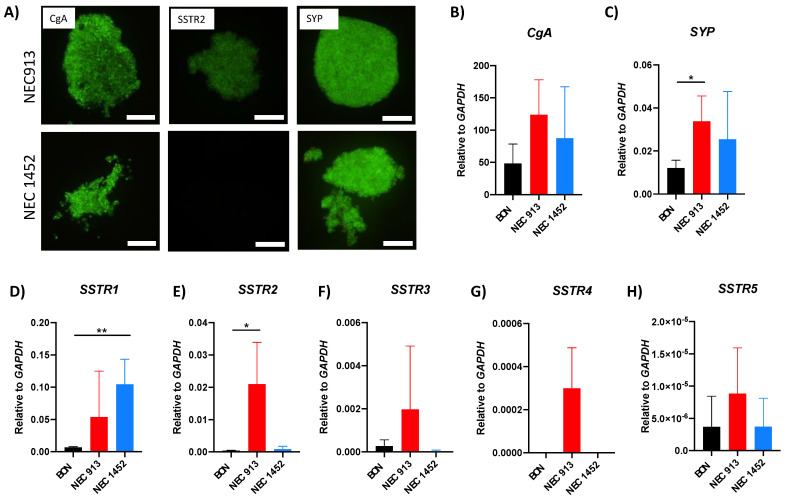
Characterization of NEC913 and NEC1452 cells for NET markers: (**A**) NEC913 and NEC1452 cells incubated with antibodies against CgA (1/300), SSTR2 (1/300), and SYP (1/600) overnight and with secondary antibodies coupled to FITC (1/500) for 1 h at room temperature. Microscopy pictures are taken using 200 ms exposure time. Scale bar represents 100 μm. (**B**–**H**) Gene expression analyses of NET markers in NEC913 and NEC1452 cells compared to BON cells. Gene expression levels were normalized to the control gene *GAPDH* to determine the relative fold change. Statistical analyses of gene expression changes were performed using T-tests in Prism GraphPad. *p* < 0.05 was depicted with *. *p* < 0.01 was depicted with **.

**Figure 4 cancers-14-01910-f004:**
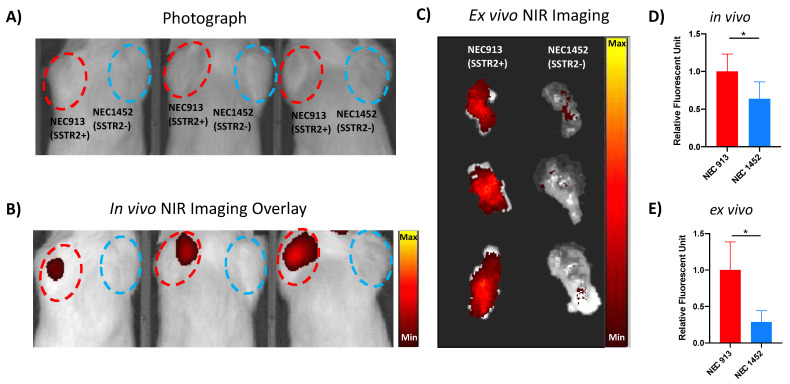
Application of NEC PDX models for SSTR2-targeted imaging: (**A**) Representative photograph of mice harboring NEC913 and NEC1452 tumors ranging from 1.0 to 2.0 cm in diameter 5 weeks post tumor cell injections from an *n* = 5 mice experiment. (**B**) Representative Near infrared (NIR) fluorescence imaging of mice harboring NEC913 and NEC1452 tumors using exposure time of 2 s and excitation and emission wavelengths set at 740 and 790 nm, respectively. NEC913 tumors are circled in red and NEC1452 tumors are circled in blue. (**C**) Representative ex vivo NIR fluorescence imaging of dissected NEC913 and NEC1452 tumors from an *n* = 5 mice experiment. (**D**,**E**) Quantifications of in vivo and ex vivo NIR fluorescence signal in NEC913 and NEC1452 tumors. T-tests were performed using Prism GraphPad. *p* < 0.05 was depicted with *.

**Figure 5 cancers-14-01910-f005:**
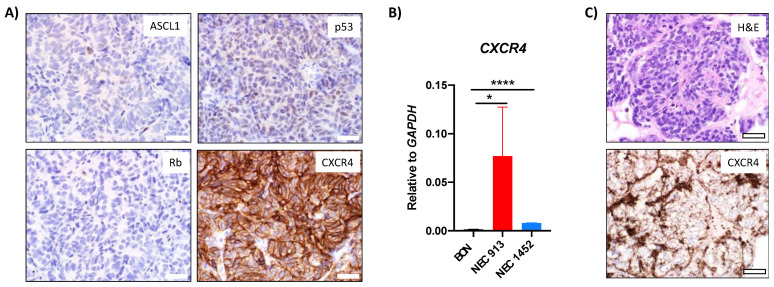
Additional characterization of NEC913 PDX model: (**A**) IHC analyses of NEC913 PDX tumors for NEC markers such as Rb, p53, ASCL1, and CXCR4. (**B**) Comparison of the gene expression levels of *CXCR4* in BON, NEC913, and NEC1452 cells by quantitative PCR normalized to the control gene *GAPDH* to determine the relative fold change. Statistical analyses of gene expression changes were performed using T-tests in Prism GraphPad. *p* < 0.05 was depicted with *. *p* < 0.0001 was depicted with ****. (**C**) NEC913 spheroids H&E staining and IHC analysis of CXCR4. Scale bar represents 40 μm.

**Table 1 cancers-14-01910-t001:** List of primer sequences used for qPCR experiments.

Gene Symbol	Forward	Reverse
*GAPDH*	GGAGCGAGATCCCTCCAAAAT	GGCTGTTGTCATACTTCTCATGG
*CGA*	TAAAGGGGATACCGAGGTGATG	TCGGAGTGTCTCAAAACATTCC
*SYP*	CTCGGCTTTGTGAAGGTGCT	CTGAGGTCACTCTCGGTCTTG
*SSTR1*	GCGCCATCCTGATCTCTTTCA	AACGTGGAGGTGACTAGGAAG
*SSTR2*	TGGCTATCCATTCCATTTGACC	AGGACTGCATTGCTTGTCAGG
*SSTR3*	AGAACCTGAGAATGCCTCCTC	GCCGCAGGACCACATAGATG
*SSTR4*	GCATGGTCGCTATCCAGTG	GCGAAGGATCACGAAGATGAC
*SSTR5*	GTGATCCTTCGCTACGCCAA	CACGGTGAGACAGAAGACGC
*CXCR4*	ACGCCACCAACAGTCAGAG	AGTCGGGAATAGTCAGCAGGA

**Table 2 cancers-14-01910-t002:** List of GEP NEN patient tumor samples used for patient-derived xenograft (PDX) development.

Patient Tumor ID Number	Classification of Tumor	WHO Terminology	Differentiation	Tumor Grade	Ki67 (%)	Establishment of PDX
PNET459	Pancreatic NET	NET Grade 2	Well differentiated	Intermediate	7	no
PNET560	Pancreatic NET	NET Grade 2	Well differentiated	Intermediate	8.4	no
PNET1164	Pancreatic NET	NET Grade 2	Well differentiated	Intermediate	13	no
SBNET1063	Small bowel NET	NET Grade 3	Well differentiated	High	80	no
NEC913	Ampullary NEC	NEC, small and large-cell types	Poorly differentiated	High	80–90	yes
NEC1452	Rectal NEC	NEC, large-cell type	Poorly differentiated	High	80–90	yes

## Data Availability

Please contact the corresponding authors for any additional data.

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
