# Peer review of "Establishment of Novel Neuroendocrine Carcinoma Patient-Derived Xenograft Models for Receptor Peptide-Targeted Therapy"

_cancers, 2022, doi:10.3390/cancers14081910_

Round 1

Reviewer 1 Report

The authors have addressed the comments and questions that I raised in my previous review report. As a result, and also following comments of the other reviewer, the manuscript has improved significantly. 

However, my main issue, the lack of experiments showing the potential of these models, has not been addressed, other then a remark by the authors, that this may take several years. Therefore, I feel this work does not represent a sufficient advancement of the field.

Reviewer 2 Report

Hereby, I recommend the manuscript for publication in Cancer Journal.

Reviewer 3 Report

The authors responded adequately to my concerns raised. Thank you.

I was surprised that the NEC cells of epithelial nature was less adhesive to culture dish than mesenchymal murine fibroblasts. This must be a welcome news to readers.

a spelling error: opitimal should be optimal

This manuscript is a resubmission of an earlier submission. The following is a list of the peer review reports and author responses from that submission.

Round 1

Reviewer 1 Report

This is an interesting and meritable manuscript on the generation of in vitro and in vivo models of neuroendocrine carcinomas of the GI tract. The work and experiments are well-described. Apart from a few conceptional items, that I will specify below, my main problem with the manuscript is that it stops where the reader would expect the authors take this work further. I am aware that the generation of these models involves extensive work and control experiments, but in order to show the validity and real potential, more work is needed.

Major comments:

  1. The concept of neuroendocrine tumors versus neuroendocrine carcinomas is only partly explained. Specifically the grading of NET, into G1, G2, and G3 is not well-explained and should be added to the introduction. In addition, table 1 should be adapted as well, since NEC do not include grading, so only NET should have a grade.
  2. Some additional explanation seems needed for the retroperitoneal NEC. In table 1 it is presented as a retroperitoneal tumor, but in the text the authors state it has been clinically classified as a primary rectal NEC. I acknowledge in this case it must have been difficult to pinpoint the primary tumor location, but if the clinical diagnosis is genuine, then this should be presented as a rectal NEC (even if with the benefit of the doubt).
  3. Reference 19 is outdated to use as a reference to current GEP-NET treatment. This can easily be replaced by one or more recent ones. It would be good to specify peptide receptor radionuclide therapy as a term (which is probably meant by radiolabeled SSAs?).
  4. Similar to question 2: is the ampullary tumor really a NEC or is it a NET G3? Looking at the histology it is difficult to see the cellular and nuclear detail but the architecture looks very much like a NET. Please specify and adapt where necessary.

Minor comments:

  1. Results, bottom of page 4: A separate portion ... was (instead of were) ...
  2. Intro, last sentence: theranostics (without the g).
  3. Page 8, line 6: detect (without s).
  4. The sentence in the discussion referring to Figures 1B and 5A.B, on page 10, is not good. Please rephrase.

Author Response

We thank Reviewer 1 for insightful and constructive comments of our manuscript.  We have addressed these comments in the Point-by point response attached. 

Reviewer 2 Report

It was a pleasure to review the paper by Tran et al. The concept of PDX is extremely difficult to solve in patients with neuroendocrine cancers. Other questions will arise when implementing experimental therapies, including patient-derived xenografts and organoids.

Just minor questions regarding this article to be clarified by the authors:

1) please include Ki67% of patients in Table 1

2) A biopsy revealed a G3 NEC of the ampulla and the patient was treated with carboplatin/etoposide chemotherapy for 4 months, then had a Whipple procedure, where a 2.5 cm primary tumor with multiple involved nodes were also removed

Chemotherapy can influence the quality of cells frozen for organoid/xenograft implementation. Please explain potential risks with chemotherapy and justify material used from the previously treated patient. How many chemotherapy cycles did the patient get, and what is the possible influence on specimen quality?

Author Response

We thank Reviewer 2 for insightful and constructive comments of our manuscript.  We have addressed these comments in the Point-by point response attached. 

Reviewer 3 Report

The authors described establishment and characterization of 2 novel GEP NEC PDX lines with their corresponding cell lines. One of them expressed SSTR2 and CXCR4, and particularly had a significant potential to be used for theranostics research for this tumor type. The manuscript is well written, organized and reporting an important issue. Lack of information was noticed and should be improved with some other minor points.

Major concerns:

  1. Methods. 2.1. Cell lines and patient-derived xenograft models: Detailed information for cryopreserved status of “six patient tumor samples” should be presented. For example, storage temperature, device, or the way of freezing (just snap frozen or frozen with cryopreservation solution) are critical. If they were just snap frozen samples, it was an issue to be emphasized.
  2. Information for establishment of cell lines for NEC913 and NEC1452 should be provided. Usually, PDX derived culture is contaminated with vivid murine fibroblasts and needed removal of murine cells with some methods.
  3. Pathologic diagnosis of NEC913 was uncertain, is it small cell NEC or large cell NEC?
  4. p53 Immunohistochemistry for NEC913 PDX tumor: Although the authors described as “confirmed low expression of the truncated p53”, this may not be accurate. They have to interpret this just as low signals with the antibody used.

Minor concerns:

  1. Writing of NETS, NETs, GEP NEC, GEP NECs or similar words should be united. Nouns were seemed to be with s and adjectives without, but there were some exceptions.
  2. Writing of human gene symbols should be in uppercase italics.
  3. Method to evaluate the PDX volume should be provided.
  4. Improvement of figure resolution would be appreciated.
  5. Words such as NIR, FDG-PET, DOTATATE PET, SBNET or PNET should be spelled out when they first appeared.

Author Response

We thank Reviewer 3 for insightful and constructive comments of our manuscript.  We have addressed these comments in the Point-by point response attached. 
